

# Enhanced information cross-attention fusion for drug–target binding affinity prediction

Ailu Fei[1], Yihan Wang[2], Tiantian Ruan[2], Yekang Zhang[1], Min Yao[2] and Li Wang[3]

[1] School of Information Science and Technology, Nantong University, Nantong, China
[2] Department of Immunology, Medical School, Nantong University, Nantong, China
[3] Research Center for Intelligent Information Technology, Nantong University, Nantong, China

## ABSTRACT

**Background:** The rapid development of artificial intelligence has permeated many fields, with its application in drug discovery becoming increasingly mature. Machine learning, particularly deep learning, has significantly improved the efficiency of drug discovery. In the core task of predicting drug–target affinity (DTA), deep learning enhances predictive performance by automatically extracting complex features from compounds and proteins.

**Methods:** Traditional approaches often rely heavily on sequence and two-dimensional structural information, overlooking critical three-dimensional and physicochemical properties. To address this, we propose a novel model—Cross Attention Fusion based on Information Enhancement for Drug–Target Affinity Prediction (CAFIE-DTA)—which incorporates protein 3D curvature and electrostatic potential information. The model approximates protein surface curvature using Delaunay triangulation, calculates total electrostatic potential *via* Adaptive Poisson-Boltzmann Solver (APBS) software, and employs cross multi-head attention to fuse physicochemical and sequence information of proteins. Simultaneously, it integrates graph-based and physicochemical features of compounds using the same attention mechanism. The resulting protein and compound vectors are concatenated for affinity prediction.

**Results:** Cross-validation and comparative evaluations on the benchmark Davis and KIBA datasets demonstrate that CAFIE-DTA outperforms existing methods. On the Davis dataset, it achieved improvements of 0.003 in confidence interval (CI) and 0.022 in $R^2$. On the KIBA dataset, it improved MSE by 0.008, CI by 0.005, and $R^2$ by 0.017. Compared to traditional models relying on 2D structures and sequence data, CAFIE-DTA shows superior performance in DTA prediction. The source code is available at: https://github.com/NTU-MedAI/CAFIE-DTA.

## INTRODUCTION

Artificial Intelligence (AI) employs anthropomorphic knowledge, learning from the solutions it generates, thus progressively enhancing its capability to tackle complex issues.

Corresponding authors
Min Yao, erbei@ntu.edu.cn
Li Wang, wangli@ntu.edu.cn

The substantial augmentation in computational capabilities, coupled with advancements in AI technology, can be harnessed to transform the drug development process fundamentally (*Mak & Pichika, 2019*).

Machine learning (ML) technology plays an increasingly important role in predicting drug–target affinity (DTA). Existing models often use the Simplified Molecular Input Line Entry System (SMILES) to represent compound structures and input features such as protein amino acid sequences, and molecular and protein secondary and tertiary structures. These features are then processed through a series of deep learning architectures, such as convolutional neural networks (CNN), recurrent neural networks (RNN), graph neural networks (GNN), Transformers, and their variants, to achieve precise predictions of DTA tasks (*Zhang et al., 2018*; *Segler et al., 2018*; *Wu et al., 2022b*). With the accumulation of protein-ligand binding data and advancements in deep learning technology, numerous machine learning and deep learning methods have emerged to predict protein-ligand binding affinity.

Utilizing natural language processing (NLP) technology to analyze the semantic characteristics of sequences, particularly in the context of small molecule compounds and protein sequence information, has become a research hotspot. Early researchers often constructed static embeddings based on one-dimensional sequence information (*e.g.*, one-hot vectors). The first deep learning DTA prediction model, DeepDTA, used SMILES to encode drug chemical structures and protein amino acid sequences to represent protein features (*Öztürk, Özgür & Ozkirimli, 2018*). The SMILES-encoded drugs, after integer labeling, along with protein sequences, were input into a CNN with three layers of one-dimensional convolution, where the drug part used max-pooling to extract potential features, and the protein part used a similar CNN structure. The feature vectors of both were then concatenated and trained through a fully connected layer for prediction. Another deep learning model, DeepAffinity, serialized SMILES-encoded drugs and structural attributes of proteins (*Karimi et al., 2019*). This model used a seq2seq autoencoder (a type of RNN) to encode SMILES and protein structure sequences into embedded representations, with seq2seq capturing internal dependencies within sequences through an Encoder-Decoder structure (*Sutskever, Vinyals & Le, 2014*).

Despite the widespread use of one-hot encoding in the above methods, this fixed mechanism has common issues. After processing small molecules and protein sequences, the resulting sub-sequence sets are large, and the vector dimensions are proportional to the number of sub-sequences, leading to dimensionality explosion, severe sparsity issues, and significantly increased computational demands. Constructing simple one-hot word vectors based solely on SMILES codes and the minimal characters of protein amino acid residues may not guarantee semantic similarity of the minimal sequence units. Therefore, there is a need to find representations that solve information loss and matrix sparsity caused by one-hot encoding.

To address this problem, researchers treat complex SMILES strings as sentences, with each atom and bond symbol regarded as words or tokens. By collecting a large *corpus* of compounds, such as treating the SMILES string "CC(=) OC1=C" as a sentence composed

of individual symbols (*Wicker et al., 2010*), models such as Extended-Connectivity Fingerprints (ECFPs) have been designed for structure-activity modeling, widely used for compound feature representation (*Rogers & Hahn, 2010*). ECFPs assign identifiers to each atom based on their characteristics, then iteratively update the identifiers according to adjacent atoms' identifiers, forming identifiers for substructures within molecules. The number of iterations determines the size of substructures the algorithm can recognize, and the final identifiers are hashed into fixed-length binary vectors. Similarly, Mol2vec, inspired by the word2vec model, learns vector representations of molecular substructures, treating molecules as sentences, with substructures identified by ECFPs (radius 0) as words in sentences. By learning embeddings of these substructures ("words"), frequently co-occurring substructures have similar embeddings (*Jaeger, Fulle & Türk, 2018*).

*Abbasi et al. (2020)* proposed a deep learning method, DeepCDA, combining convolutional layers and long short-term memory layers to learn new representations of local substructures from protein and compound sequences. Additionally, they proposed a bilateral attention mechanism to encode the interaction strength between protein and compound substructures (*Abbasi et al., 2020*). *Nguyen et al. (2021)* represented drugs as graphs and used graph neural networks to predict drug–target affinity in their model GraphDTA. *Jiang et al. (2022)* proposed a sequence-based weighted graph neural network prediction method, WGNN-DTA, where the weighted protein graph construction method provides more detailed residue interaction information and uses an evolutionary scale model (ESM) to significantly improve computational speed while maintaining accuracy. *Monteiro, Oliveira & Arrais (2022)* proposed a Transformer-based model for predicting drug–target binding affinity, DTITR, which provides an end-to-end solution. This model uses self-attention layers to capture the biological and chemical context presented in protein sequence and compound structure data (*Monteiro, Oliveira & Arrais, 2022*). *Huang et al. (2020)* proposed DeepPurpose, a model that implements 15 compound and protein encoders and over 50 neural architectures. *Zhang, Wang & Chen (2022)* proposed the MRBDTA model, consisting of embedding and position encoding, molecular representation modules, and interaction learning modules. They developed the Trans block by improving the transformer's encoder to extract molecular features (*Zhang, Wang & Chen, 2022*). In addition, *Zong et al. (2019)* proposed a network-based prediction framework that generates knowledge and modularizes feature selection and association prediction, making it easy to adapt and extend to other feature sources or machine learning algorithms. *Ma et al. (2023)* proposed to enrich protein expression by pre-training VGAE to construct protein interaction networks and similarity networks, thereby providing extensive prior knowledge in DTA prediction.

Despite significant progress made by existing deep learning models in DTA prediction, most rely solely on one-dimensional sequence information or two-dimensional graph information. Compared to molecular graphs, molecular sequences can gather sequence information for feature learning but overlook the structural information contained in molecular graphs, affecting prediction accuracy. Drug feature sequences are currently replaced by molecular graphs, which contain rich feature information. However, many

target proteins lack three-dimensional structural information, preventing the use of their high-dimensional characteristics (*Zhang et al., 2023*). So in order to compensate for the lack of three-dimensional structure, *Voitsitskyi et al. (2023)* proposed 3DProtDTA, which uses AlphaFold to predict the structure of proteins, and uses these structures to represent proteins for drug target affinity prediction. *Liu et al. (2025)* introduced a Subpocket modeling module to provide level based information for each pocket, and decomposed each pocket into sub pockets to obtain information in 3D space.

Additionally, the physicochemical properties of drugs and targets are also crucial for binding affinity prediction. Ignoring these properties may lead to incomplete features and limited model performance. Especially the electrostatic potential energy and surface curvature of protein surfaces. *Bitencourt-Ferreira & de Azevedo Junior (2021)* reviewed the development of scoring functions and proposed that electrostatic potential has a significant impact on protein drug binding affinity by applying semi empirical free energy scoring functions to predict binding affinity. *Li et al. (2023)* proposed that surface area and dihedral angle, important components of protein three-dimensional structure, play a crucial role in predicting protein binding sites. The surface area and dihedral angle of proteins can be reflected in the surface curvature of proteins, which can effectively help find the binding site. At the same time, the accuracy of the binding site directly affects the binding affinity between proteins and drugs.

In order to overcome the limitations of target proteins in lacking three-dimensional structural information, physical and chemical properties, and information fusion, we propose the CAFIE-DTA model. This model innovatively integrates the three-dimensional structural information of proteins (such as protein surface curvature and electrostatic potential energy) with sequence information through attention mechanism, as well as the physical and chemical information of drugs with graph information through attention mechanism. First, we independently obtain the physicochemical information of drugs and proteins and represent it as matrices. Then, features are extracted from the graph constructed from drug SMILES using a dual GNN with residual connections, and drug graph features are fused with the corresponding physicochemical information features through a multi-head cross-attention mechanism. Similarly, the protein features extracted by CP-Encoder are fused with the corresponding physicochemical information through a multi-head cross-attention mechanism. Finally, the fused features are concatenated and input into a fully connected layer to predict affinity values. Cross-validation and comparative evaluations with existing methods on the benchmark datasets Davis and KIBA showed an improvement of 0.003 in CI and 0.022 in $R^2$ on the Davis dataset, and an improvement of 0.008 in MSE and 0.005 in CI and 0.017 in $R^2$ on the KIBA dataset. CAFIE-DTA demonstrates superior performance in DTA prediction compared to traditional models that rely on 2D structures and sequence information.

## MATERIALS AND METHODS

The resource codes and datasets are available at https://github.com/NTU-MedAI/CAFIE-DTA.

**Table 1 Statistics of the two datasets.**

| Number | Dataset | Protein | Compounds | Binding entities |
|---|---|---|---|---|
| 1 | Davis | 442 | 68 | 30,056 |
| 2 | KIBA | 229 | 2,111 | 118,254 |

The resource Davis and KIBA datasets can be downloaded at staff.cs.utu.fi/~aatapa/data/DrugTarget/.

## Benchmark datasets

To compare CAFIE-DTA with existing machine learning and deep learning-based models, the model was evaluated on two publicly available DTA datasets: the Davis dataset and the KIBA dataset (*Davis et al., 2011*; *Tang et al., 2014*). The Davis dataset consists of 442 proteins and 68 compounds, forming 30,056 drug–target pairs, with the kinase dissociation constant (Kd) values used as the measure of binding affinity. Higher Kd values indicate lower binding strength between the drug and the target. The KIBA dataset uses KIBA scores as the measure for predicting drug–target affinity, where higher KIBA scores indicate stronger binding between the drug and the target. The KIBA dataset comprises 229 proteins and 2,111 compounds, forming 118,254 drug–target pairs. In both datasets, drug SMILES strings were collected from the PubChem compound database, and protein sequences were collected from the UniProt protein database. Table 1 provides the statistics of these two datasets. It should be noted that due to computer memory limitations, one long protein sequence and its related pairs were removed from the KIBA dataset.

The Davis dataset comprises 442 kinase proteins and their corresponding inhibitors (68 ligands), each with a dissociation constant value. The Kd values were transformed into the logarithmic scale as pKd, serving as the binding affinity values.

$$pK_d = -log10\left(\frac{K_d}{1e9}\right).$$

The KIBA dataset is developed based on the KIBA method, initially comprising 467 proteins, 52,498 drugs, and their binding affinity scores. Here, the KIBA score measures the bioactivity of kinase inhibitors and is regarded as a binding affinity value. After filtering by SimBoost (*He et al., 2017*), it contains 229 unique proteins and 2,111 unique drugs for a fair comparison. Regarding the input of proteins and drugs in the Davis and KIBA datasets, we follow the DeepDTA approach and digitize protein sequences into a fixed maximum length through a dictionary (*Öztürk, Özgür & Ozkirimli, 2018*).

## Physical and chemical information acquisition of compound and proteins

For each drug, the physicochemical information is obtained from the PubChem official website, including lipophilicity, topological polar surface area, complexity, number of hydrogen bond donors, number of hydrogen bond acceptors, number of heavy atoms, total charge of the molecule, and specified atomic counts (such as 'C', 'H', 'N', 'O', 'F', 'S', 'Cl', 'Br', 'I'). The physicochemical properties of a given protein are calculated using functions

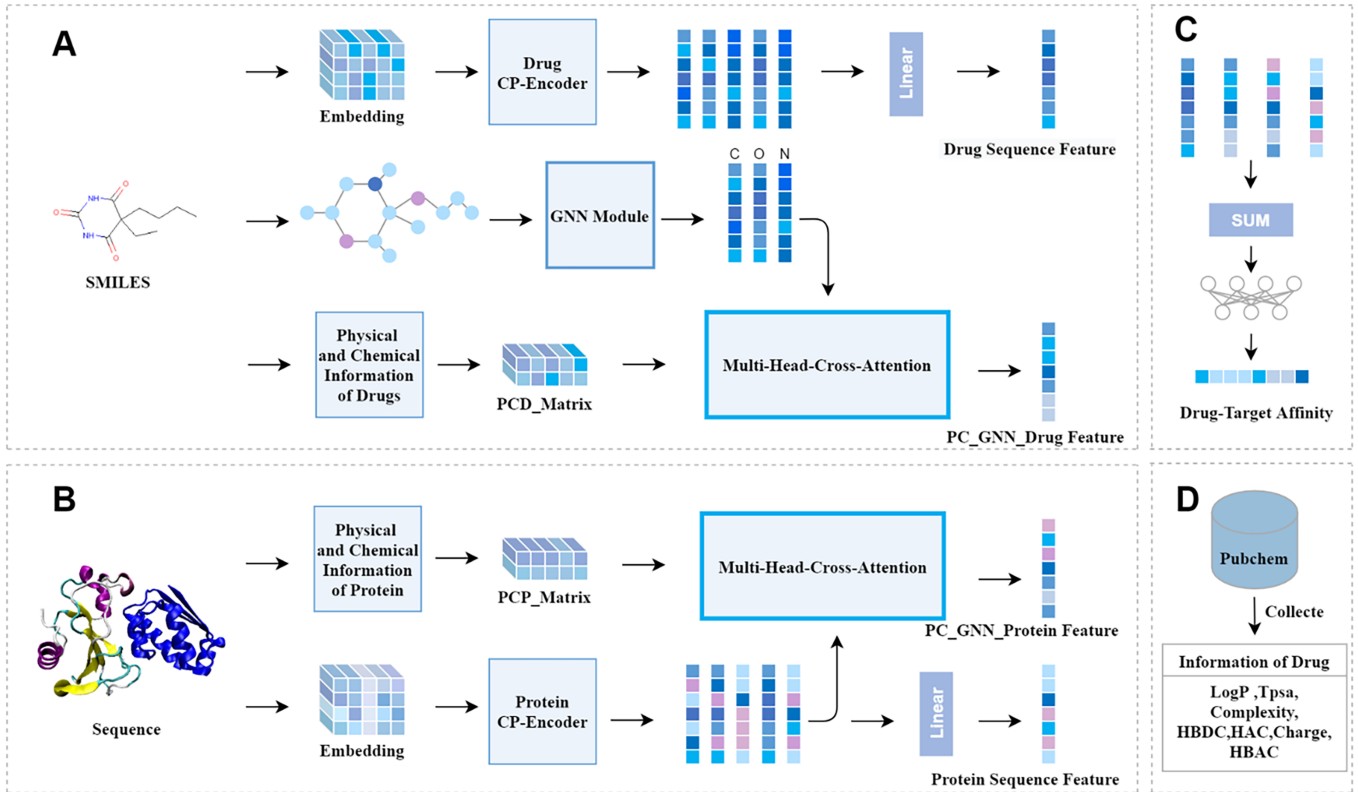

**Figure 1 The overall architecture of CAFIE-DTA.** (A) The drug information feature acquisition module takes SMILES molecular formula as its input and outputs drug related feature vectors. This model is responsible for obtaining the sequence features of drugs and features with two-dimensional structure information and physicochemical information. (B) The protein information feature acquisition module takes protein sequences as input and protein related feature vectors as output. This model is responsible for obtaining the sequence features and physicochemical information of proteins. (C) Affinity prediction module. This architecture is responsible for combining feature vectors from both drug and protein aspects, and inputting them into a multi-layer perceptron to obtain specific affinity values. (D) This module mainly introduces the sources of specific values of drug physicochemical information and the characteristic information included. PCD represents the physicochemical information of drugs, PCP represents the physicochemical information of proteins, and CP represents compounds and proteins.

within Biopython to compute hydrophobicity indices of the protein's amino acids, its molecular weight, and isoelectric point. APBS software is used to calculate the total electrostatic potential of proteins. Using Delaunay triangulation and circumcircle radius to estimate the average surface curvature of proteins. The specific method for obtaining the physical and chemical characteristics of drugs is to use Python to call the PubChempy interface on the official PubChem website to obtain the physical and chemical characteristics. Write the physical and chemical information features obtained through the interface into a list one by one to obtain a vector form, and sometimes some values may be missing in the physical and chemical information features of these drugs. To address these missing values, set it to 0. For other values, in order to preserve their original information without excessive processing, they are directly represented as raw values.

## The overview of the proposed framework

Figure 1 depicts our model. It can be seen that the model is mainly composed of three modules: the drug feature extraction module, the protein feature extraction module, and the prediction module. The drug feature extraction module in Fig. 1A, the physicochemical information matrix is fused with the features of the molecular graph through a multi-head cross-attention mechanism. The CP-Encoder is used to extract the sequence information of the drug. The protein feature extraction module in Fig. 1B, the physicochemical information matrix is obtained by concatenating the physicochemical information of the protein, and the sequence features are extracted using the same CP-Encoder as in the drug feature extraction module. On one hand, the sequence information is fused with the physicochemical information matrix through a multi-head cross-attention mechanism; on the other hand, the sequence information features are input separately. The prediction module in Fig. 1C, the prediction module consists of fully connected layers. The features obtained from the drug feature extraction module and the protein feature extraction module are concatenated and input into the prediction module. The physicochemical information of the corresponding drug is obtained from the PubChem website, as shown in Fig. 1D, and the number of specific atoms is calculated. The values of this physicochemical information are concatenated into a one-dimensional vector, and the same processing is done for each drug. These one-dimensional vectors are then combined into a physicochemical information matrix.

## Drug physical and chemical information matrix

The physicochemical information of the corresponding drug is obtained from the PubChem website. This physicochemical information includes lipophilicity, topological molecular polar surface area, complexity, number of hydrogen bond donors, number of hydrogen bond acceptors, number of heavy atoms, and the total charge of the molecule. Additionally, it includes the number of specified atoms. The values corresponding to this physicochemical information are concatenated into a one-dimensional vector $E_d$. Similarly, the physicochemical information corresponding to the remaining drugs can be obtained. Finally, this physicochemical information is combined into a physicochemical information matrix $A_d^{pc}$.

## Molecular graph feature extraction

When directly represented as strings, structural information of molecules is lost. Consequently, drugs are represented as molecular graphs to encompass more features. We employ RDKit to convert SMILES codes into corresponding molecular graphs and extract atomic features. Graph neural networks (GNNs) can effectively leverage the spatial topological structure information of the graph to extract latent features, aggregating node features to attain a graph-level representation. The following provides a brief overview of graph convolutional networks (GCNs). For a molecular graph $G = (V, E)$, where $V$ denotes the set of nodes and $E$ the set of edges, each atom's initial feature vector is denoted as $Xi$. The graph comprises a feature matrix $X \in R(N * M)$ and an adjacency matrix $A \in R(N * N)$, with $N$ representing the number of nodes and $M$ the dimensionality of the

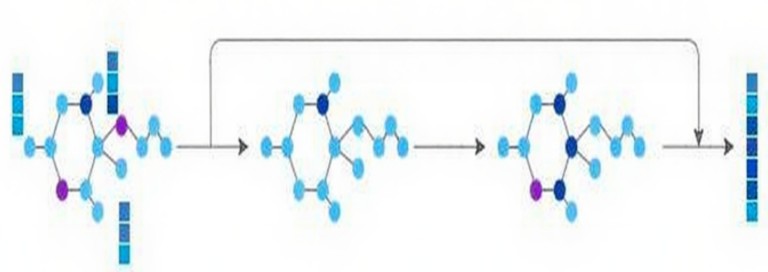

**Figure 2 GNN module.**

features. The adjacency matrix represents the interaction relationships between nodes. To describe the propagation mechanism of a GCN layer, it is defined as

$$H^{(l+1)} = \sigma\left(\tilde{D}^{-\frac{1}{2}}\tilde{A}\tilde{D}^{-\frac{1}{2}}H^{(l)}W^{(l)}\right). \tag{1}$$

Here, $\tilde{A}$ signifies the adjacency matrix with incorporated self-loops, $\tilde{D}$ symbolizes the degree matrix of the graph, $H(i)$ denotes the feature matrix at layer $i$, $H^{(l+1)}$ indicates the feature representation following $l$ iterations of message passing, $\sigma$ represents the rectified linear unit (ReLU) activation function, and $W^{(l)}$ denotes the trainable weight matrix for at the layer $i$. The feature matrix $X$ serves as the input to $H^{(0)}$.

Concurrently, to mine structural information, the molecular graph is channeled into the GNN module, comprised of two successive GNN layers. Residual connections are adopted to combine the initial input with the outputs from these two GNN layers, thereby preserving the original molecular graph information. Ultimately, the characteristics of the molecular graph are derived, as depicted in Fig. 2.

## Drug sequence feature extraction

The drug sequence feature extraction module primarily consists of the CP-Encoder, which itself is comprised of position embeddings, input embeddings, and a Transformer Encoder. Here, the definition formula for drug $D$ is given as Eq. (2).

$$D = \{d_1, d_2, d_3, \ldots\ldots d_z, \ldots\ldots, d_{dl}\}\ d_z \in N_d, \tag{2}$$

where $d_i$ denotes the character at layer $i$ in the SMILES string, $N_d$ represents a collection of 62 SMILES characters, and $d_1$ signifies the length of the SMILES representation for drug $D$. To incorporate the relative or absolute positional information of each atom within drug $D$, position embeddings are utilized. A superparameter $k$ is defined to represent the maximum length of a drug. The position embeddings are denoted as $PE^D \in R^{(k*t)}$, with the concrete embedding techniques outlined by Eqs. (3) and (4) respectively.

$$PE^D(i, 2j) = \sin\left(\frac{i}{10000^{\frac{2j}{t}}}\right)\quad i = 1, 2, 3, \ldots\ldots, n_D(n_D \leq k) \tag{3}$$

$$PE^D(i, 2j+1) = \cos\left(\frac{i}{10000^{\frac{2j}{t}}}\right)\quad j = 0, 1, 2, 3, \ldots\ldots, \frac{2}{t}, \tag{4}$$

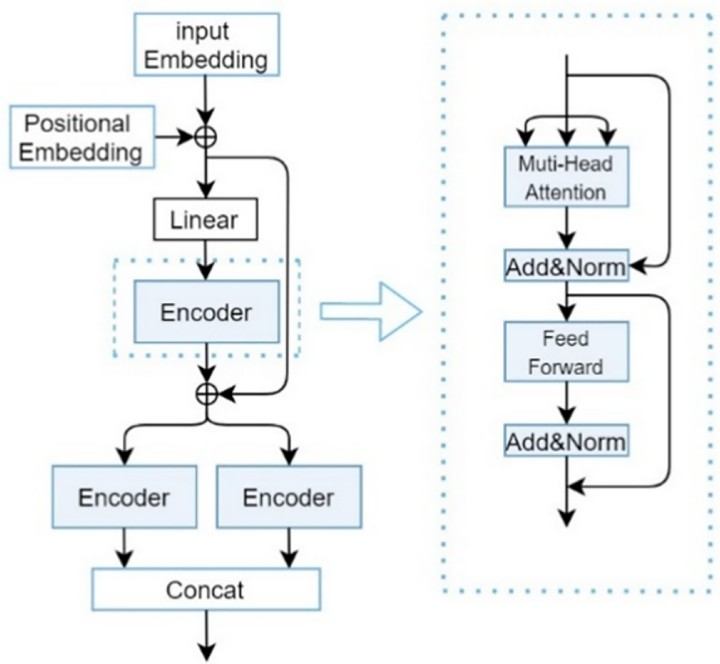

**Figure 3  CP-encoder module.**               

where $t$ signifies the dimension size of the positional embedding for the drug SMILES. $PE^D(i, :)$ refers to the row of matrix $PE^D$ at layer $i$, representing the positional embedding for the atom $i$ in the drug. Equation (5) is defined to express the output that combines input embeddings and positional embeddings:

$$X^D = E^D + PE^D, X^D \in R^{k*t}, \tag{5}$$

The transformer encoder primarily comprises multi-head attention mechanisms. The attention weight matrix is illustrated as in Eq. (6).

$$Attention(Q, K, V) = softmax\left(\frac{QK^T}{\sqrt{d_k}}\right) * V. \tag{6}$$

Here, the inputs for Q = K = V are $X^D$. The $head_i$ is defined by Eq. (7).

$$head_i = Attention\left(QW_i^Q, KW_i^K, VW_i^V\right), \tag{7}$$

where $QW_i^Q, KW_i^K, VW_i^V$ represent the linear projection matrices for query (Q), key (K) and value (V), respectively. Finally, the outputs of the four scaled dot-product attention layers are concatenated and passed through a linear layer to yield the MultiHead (Q, K, V) output of the multi-head attention layer. The multi-head (Q, K, V) is defined by Eq. (8).

$$MultiHead(Q, K, V) = Concat(head_1, \ldots\ldots, head_h)W^O, \tag{8}$$

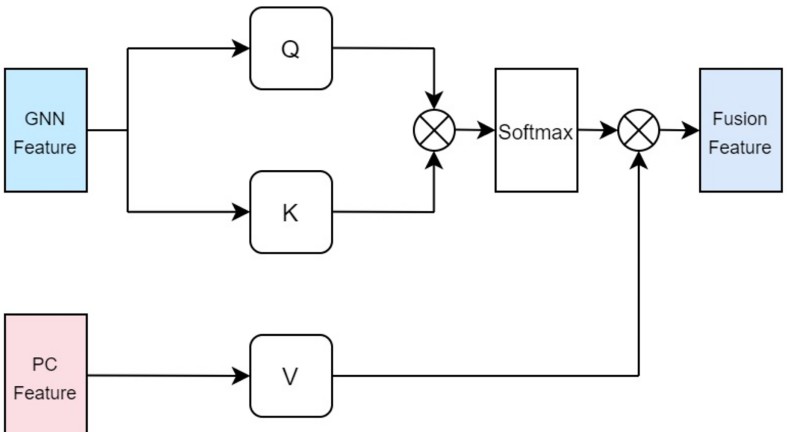

**Figure 4  Cross attention module of drugs.**

$W^O$ denotes a linear projection matrix. Here, by taking $X^D$ as the input for Q, K, and V, we obtain the sequence features of drugs, denoted as $F_D^T$. The composition of the CP-Encoder is inspired by the encoding module proposed by *Zhang, Wang & Chen (2022)*, with its specific structure illustrated in Fig. 3.

Although traditional Transformers can be used to extract one-dimensional sequence information and increasing the depth (*i.e.*, layers) of Transformers can usually improve model performance, overly deep networks can also bring problems such as increased training difficulty and high computational resource consumption. At the same time, due to the long length of SMILES sequences and protein sequences, and the importance of the relationships between each atom, a single Transformer is difficult to extract deeper features and more accurate relationships between atoms. So after a series of experiments and optimizations, we found that the three-layer encoder structure can maintain a relatively low computational cost while ensuring model performance.

The main responsibility of the first encoder is to receive initial input data and begin preliminary feature extraction. The second encoder will receive preliminary processed data (including the original input information transmitted through residual connections) based on the first encoder. This layer may focus on refining and deepening feature representations to gain a deeper understanding of complex patterns in input data. Due to the different parameters of the two parallel encoders, the collected feature information is different, so adding them together can obtain more information. Secondly, it can greatly enhance the robustness of our deep learning models.

## The multi head cross attention mechanism of drugs and protein

In our approach, the drug multi-head cross-attention mechanism focuses on modeling various levels of interaction information between the molecular graph based on drugs and the physicochemical information features of drugs in high-dimensional space, generating features of different interaction levels. The formulas are identical to the aforementioned Eqs. (5), (6), (7), with the distinction lying in the difference of inputs. Specifically, the

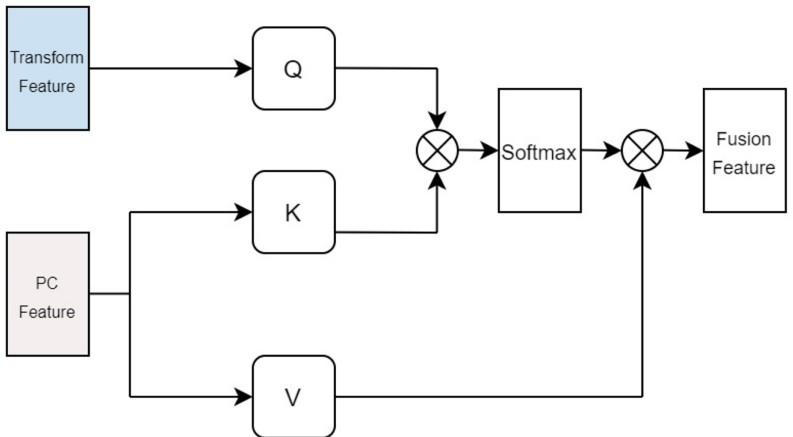

**Figure 5 Cross attention module of proteins.**

query Q employs the computed physicochemical information matrix $A_d^{pc}$ as input, while the keys K and values V utilize the molecular graph features extracted by GCN. This multi-head cross-attention mechanism, through this particular input methodology, identifies parts that are most relevant to both the structural characteristics of the drugs and their physicochemical properties. Ultimately, the fused feature is denoted as illustrated in Fig. 4.

In terms of protein feature fusion, a module identical to the multi-head attention mechanism for drugs is employed to integrate the physicochemical information and sequence information of the protein. Here, the query Q takes the computed matrix of physicochemical information $A_p^{pc}$ as input, while the keys K and values V utilize the sequence features extracted by the Transformer as inputs. Cross multi-head attention is used to fuse the physicochemical information and sequence feature information, thereby allowing for simultaneous consideration of the protein's sequential information alongside its physicochemical properties, preventing the loss of valuable information. The fused features are denoted as illustrated in Fig. 5.

## Physical and chemical information matrix of proteins

The physicochemical properties of proteins are calculated using functions within Biopython to determine hydrophobicity indices of amino acids, the molecular weight, and the isoelectric point of the protein. Additionally, the total electrostatic potential of the protein is computed using the APBS software. This process begins with downloading the corresponding PDB file for the protein in question, which is then fed into the APBS software to calculate its overall electrostatic potential. Regarding the calculation of mean curvature, it is approached through discrete mathematical approximations and geometric analyses, specifically employing concepts of Delaunay triangulation and circumradius (*Zhou & Yan, 2014*). The protein molecule's surface is approximated by a set of interconnected triangles (or tetrahedra), constituting a mesh derived *via* Delaunay triangulation. Delaunay triangulation is a method for constructing interconnected

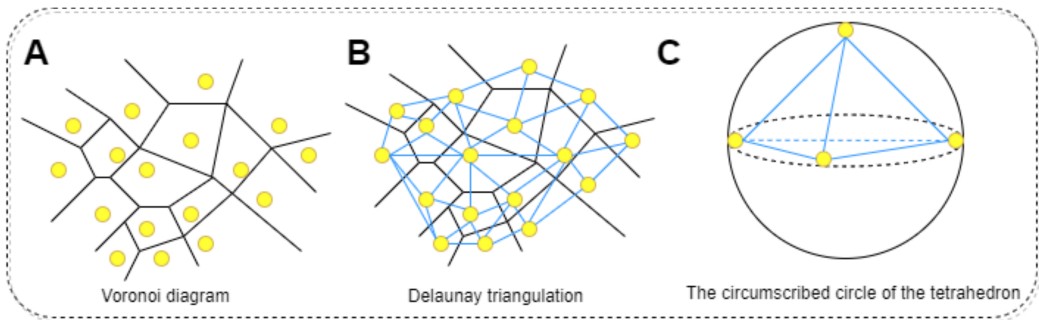

**Figure 6** **The process of calculating the radius of the circumscribed circle through triangulation.** (A) Voronoi diagram for a set of points. Each point represents the residue of the protein. (B) Delaunay triangulation can be obtained by connecting all the points that share common Voronoi faces. The edges of Delaunay triangulation (bold segments) represent the connection network of the points. (C) Calculating the coordinates of each vertex of a tetrahedron, the length of each edge, and the volume. Constructing the outer tangent sphere of a tetrahedron through vertex coordinates. Calculate curvature through this circumcircle.

triangles over a given set of points, with each point representing the coordinate of a protein atom. This step facilitates the construction of local approximations of the surface. For each generated tetrahedron, the circumradius (R) is calculated. The circumcircle is an externally tangent sphere constructed through the vertices of the tetrahedron. Curvature is inferred from this circumradius, conventionally defined as 1/R; higher curvature indicates a more recessed surface, while lower curvature suggests a more protruding one. To find the average curvature, the curvatures of all tetrahedra are summed and then divided by the count of valid tetrahedra, those with non-zero volume. In summary, this method harnesses mathematical approximations and geometric analyses to decompose the protein surface through Delaunay triangulation into a set of tetrahedra, with curvature computed at each tetrahedron. By averaging these curvature values, the mean curvature of the entire protein molecular surface can be obtained. The procedure for calculating the surface curvature of proteins involves: obtaining the PDB files for each protein in the dataset, using these PDB files as input to retrieve the three-dimensional coordinates of the protein. Applying the Delaunay function to perform Delaunay triangulation, thereby creating a triangular mesh approximating the protein molecular surface. Converting this triangular mesh into tetrahedra, followed by computing the coordinates of each vertex, lengths of edges, and volumes. Constructing the circumscribed sphere around each tetrahedron using vertex coordinates, and using this circumcircle to determine curvature. Finally, summing the curvatures of all tetrahedra and dividing by the number of valid tetrahedra yields the mean curvature. The specific process is shown in Fig. 6. All one-dimensional physicochemical property vectors are combined to form a physicochemical information matrix $A_p^{pc}$. In Fig. 6A, each yellow dot represents an amino acid residue of the protein, and the black edges represent the edges that make up the Tyson polygon. Its characteristic is that the distance from any point within a Thiessen polygon to the control points that make up the polygon is less than the distance to the control points of other polygons. Then connect all

amino acid residues according to Fig. 6B to form a tetrahedron. Finally, calculate the coordinates of each vertex, length, and volume of each edge of the tetrahedron based on Fig. 6C. Construct an outer tangent sphere of a tetrahedron using vertex coordinates. Calculate the curvature passing through the circumcircle.

The specific steps for calculating the average curvature are as follows,

Firstly, extract the three-dimensional coordinates $(x_i, y_i, z_i)$ of all atoms in the protein to form a point set P. Then, using the three-dimensional Delaunay triangulation method, divide $P$ into a series of non overlapping tetrahedral units. The characteristic of Delaunay partitioning is that the outer sphere of any tetrahedron does not contain any other input points, thus ensuring the local optimality and stability of the partitioning structure. Based on the obtained tetrahedral elements, further calculate the local geometric curvature to evaluate the overall curvature of the protein surface.

Assuming a tetrahedron has four vertices, denoted as $p_1, p_2, p_3, p_4$, their coordinates are three-dimensional vectors:

$$p_i = (x_i, y_i, z_i), \quad i = 1, 2, 3, 4. \tag{9}$$

The length of each edge of a tetrahedron is defined as:

$$
\begin{aligned}
a &= \|p_1 - p_2\| \\
b &= \|p_1 - p_3\| \\
c &= \|p_1 - p_4\| \\
d &= \|p_2 - p_3\| \\
e &= \|p_2 - p_4\| \\
f &= \|p_3 - p_4\|
\end{aligned}
$$

where $\| \cdot \|$ represents Euclidean distance (L2 norm). Then define the quantity $V$ as:

$$V = adf + (bef) + (cde) - (ae^2) - (bd^2) - (cf^2). \tag{10}$$

Define the radius of curvature $R$ of a tetrahedral circumcircle:

$$R = \frac{a^2(e^2 + f^2 - d^2) + b^2(d^2 + f^2 - e^2) + c^2(d^2 + e^2 - f^2)}{16V}. \tag{11}$$

If $V = 0$, it is considered that this tetrahedron is invalid, and the calculation is skipped. The circumcircle radius of the next tetrahedron is recalculated. For each effective tetrahedron, its contribution to local curvature is defined as:

$$k = \frac{1}{R}. \tag{12}$$

Finally, the overall average curvature $\bar{k}$ is:

$$\bar{k} = \frac{1}{N} \sum_{i=1}^{N} k_i \tag{13}$$

where $N$ is the number of all valid (*i.e.* $V \neq 0$) tetrahedra.

## Protein sequence feature extraction

Similar to the definition of drug D, $P$ is the mathematical expression of the drug, defined by Eq. (14).

$$P = \{p_1, p_2, p_3, \ldots\ldots, p_z, \ldots\ldots, p_{pl}\} p_z \in N^P, \tag{14}$$

where $N^P$ denotes the set comprising the 25 common amino acids, and the sequence length $pl$ varies depending on the protein. The hyperparameter e is also defined to represent the maximum length of a protein. $X^P$ represents the output of protein $P$ after being processed by embedding and position encoding, as defined by Eq. (15).

$$X^P = E^P + PE^P, X^P \in R^{e*u}, \tag{15}$$

u signifies the embedding size for the protein sequence, where amino acids and SMILES strings share the same embedding size (t = u). $E^P$ denotes the output of embeddings for all strings within protein $P$. $PE^P$ represents the output of position encodings for all strings in protein $P$. The embeddings and position embeddings for the strings of the protein are obtained, subsequently processed in a manner consistent with drug sequence feature extraction.

## Prediction module

The outputs from the four feature extraction components the physicochemical characteristics of proteins, the sequence characteristics of proteins, the sequence characteristics of drugs, and the physicochemical molecular graph features of drugs are concatenated to form a 512-dimensional feature vector. This vector is then fed into an FNN to generate the final output. As depicted in Fig. 1C, the FNN consists of two fully connected (FC) layers and one output layer. Each of the two FC layers is composed of 2,048 and 768 neurons respectively, while the output layer, made up of a single neuron, produces the predicted protein-ligand binding affinity.

## Experimental setup

The CAFIE-DTA model was trained using an Adam optimizer with an initial learning rate of 0.001. This implementation is done using PyTorch. Set the batch size to 32, allowing CAFIE-DTA to run up to 400 epochs. All code development was conducted on Ubuntu servers equipped with NVIDIA GeForce A6000 GPUs.

The Adam optimizer is an optimization algorithm based on gradient descent, which uses first-order moment estimation (*i.e.*, the mean of the gradient) and second-order moment estimation (*i.e.*, the uncardized variance of the gradient) to dynamically adjust the learning rate of each parameter (*Kingma & Ba, 2014*). And learning rate is a key factor in adjusting model parameters. A higher learning rate may lead to faster convergence, but it may exceed the minimum value; A lower learning rate ensures more stable convergence, but may get stuck in local minima or require longer convergence time. So after adjusting the learning rate, the final choice of 0.001 resulted in the best convergence effect and performance of the model. The choice of EPOCH is also important, as excessive EPOCH

**Table 2 Comparison results with baselines on the Davis dataset.**

| | Method | MSE | CI | $R^2$ |
|---|---|---|---|---|
| 1 | GraphDTA (GIN) (*Nguyen et al., 2021*) | 0.258 | 0.878 | 0.659 |
| 2 | GraphDTA (GAT) (*Nguyen et al., 2021*) | 0.278 | 0.866 | 0.624 |
| 3 | GraphDTA (GAT+GCN) (*Nguyen et al., 2021*) | 0.283 | 0.863 | 0.624 |
| 4 | DeepCDA (*Abbasi et al., 2020*) | 0.251 | 0.892 | 0.651 |
| 5 | WGNN-DTA (GCN) (*Jiang et al., 2022*) | 0.214 | 0.892 | – |
| 6 | WGNN-DTA (GAT) (*Jiang et al., 2022*) | 0.215 | 0.893 | – |
| 7 | DTITR (*Monteiro, Oliveira & Arrais, 2022*) | 0.224 | 0.872 | – |
| 8 | DeepPurpose (*Huang et al., 2020*) | 0.228 | 0.822 | – |
| 9 | MRBDTA (*Zhang, Wang & Chen, 2022*) | **0.206** | 0.891 (±0.007) | 0.704 (±0.04) |
| 10 | CAFIE-DTA | 0.223 | **0.896** | **0.726** |

Note:
The best performing data is highlighted in bold, and the second best results are underlined.

can lead to overfitting of the model and a decrease in its effectiveness. A too small EPOCH can lead to underfitting of the model, resulting in suboptimal performance.

# RESULTS

## Evaluation

In this work, we employed three metrics to evaluate and compare the predictive performance of CAFIE-DTA with existing methods, including the Concordance Index (CI), Mean Squared Error (MSE), and $R^2$ score (*Gönen & Heller, 2005*; *Pratim Roy et al., 2009*; *Roy et al., 2013*). The CI measures the probability of correctly predicting the relative order of randomly selected drug–target pairs based on their predicted *vs* true binding affinities, serving as an indicator of the model's fitting quality. The $r_m^2$ index is utilized to describe the likelihood of an acceptable model. The MSE (Mean Squared Error) evaluates the prediction accuracy of the model, that is, the discrepancy between predicted and actual values.

## Performance and analysis on different datasets

We use five fold cross validation and training set here, with the test set divided in a 5:1 ratio. To ensure fairness in the comparison, the model dataset being compared is consistent with the partitioning method described in this article.

The data in this study was obtained through replication. The following tables fully present the experimental results, with the best-performing data highlighted in bold and the second-best results underlined.

The CAFIE-DTA model shows advantages in several key performance metrics. Specifically, it improved the CI value by 0.003 and the $R^2$ value by 0.022 on the Davis dataset, and improved the MSE value by 0.008 and the CI value by 0.005 and the $R^2$ value by 0.017 on the KIBA dataset.

Compared to nine advanced computational models, the performance of CAFIE-DTA on the Davis and KIBA datasets is almost always the best, as shown in Tables 2 and 3.

**Table 3 Comparison results with baselines on the KIBA dataset.**

| | Method | MSE | CI | R$^2$ |
|---|---|---|---|---|
| 1 | GraphDTA (GIN) (*Nguyen et al., 2021*) | 0.164 | 0.874 | 0.762 |
| 2 | GraphDTA (GAT) (*Nguyen et al., 2021*) | 0.207 | 0.849 | 0.665 |
| 3 | GraphDTA (GAT+GCN) (*Nguyen et al., 2021*) | 0.155 | 0.880 | 0.767 |
| 4 | DeepCDA (*Abbasi et al., 2020*) | 0.177 | 0.889 | 0.680 |
| 5 | WGNN-DTA (GCN) (*Jiang et al., 2022*) | 0.149 | 0.892 | – |
| 6 | WGNN-DTA (GAT) (*Jiang et al., 2022*) | 0.155 | 0.884 | – |
| 7 | DTITR (*Monteiro, Oliveira & Arrais, 2022*) | 0.205 | 0.861 | – |
| 8 | DeepPurpose (*Huang et al., 2020*) | 0.190 | 0.831 | – |
| 9 | MRBDTA (*Zhang, Wang & Chen, 2022*) | 0.153 | 0.889 (±0.001) | 0.774 (±0.007) |
| 10 | CAFIE-DTA | **0.141** | **0.897** | **0.791** |

Note:
The best performing data is highlighted in bold, and the second best results are underlined.

**Table 4 Results predicted by CAFIE-DTA on test set of Davis and KIBA datasets for five times.**

| Dataset | TIME | 1 | 2 | 3 | 4 | 5 | Average of five times |
|---|---|---|---|---|---|---|---|
| Davis | MSE (SD) | 0.220 | 0.224 | 0.223 | 0.221 | 0.229 | 0.223 (0.003) |
| | CI (SD) | 0.897 | 0.897 | 0.895 | 0.895 | 0.894 | 0.896 (0.001) |
| | R$^2$ (SD) | 0.732 | 0.733 | 0.726 | 0.716 | 0.723 | 0.726 (0.006) |
| KIBA | MSE (SD) | 0.143 | 0.143 | 0.141 | 0.141 | 0.140 | 0.141 (0.001) |
| | CI (SD) | 0.896 | 0.896 | 0.897 | 0.895 | 0.899 | 0.897 (0.001) |
| | R$^2$ (SD) | 0.786 | 0.792 | 0.791 | 0.789 | 0.795 | 0.791 (0.003) |

**Table 5 Results of ablation experiments on the Davis and KIBA dataset.**

| Dataset | Method | MSE | CI | R$^2$ |
|---|---|---|---|---|
| Davis | -w/o Protein_PC information | 0.228 | 0.881 | 0.699 |
| | -w/o Drug_PC information | 0.232 | 0.889 | 0.697 |
| | -w/o MCAT | 0.229 | 0.886 | 0.716 |
| | -w/o GNN | 0.225 | 0.885 | 0.719 |
| | -w/o CP-Encoder | 0.36 | 0.863 | 0.638 |
| | -w/o protein surface curvature | 0.229 | 0.879 | 0.690 |
| | -w/o protein electrostatic potential | 0.230 | 0.877 | 0.695 |
| | CAFIE-DTA | **0.223** | **0.896** | **0.726** |
| KIBA | -w/o Protein_PC information | 0.174 | 0.885 | 0.755 |
| | -w/o Drug_PC information | 0.181 | 0.890 | 0.780 |
| | -w/o MCAT | 0.175 | 0.885 | 0.779 |
| | -w/o GNN | 0.174 | 0.885 | 0.781 |
| | -w/o CP-Encoder | 0.191 | 0.870 | 0.699 |
| | -w/o protein surface curvature | 0.149 | 0.892 | 0.780 |
| | -w/o protein electrostatic potential | 0.151 | 0.890 | 0.786 |
| | CAFIE-DTA | **0.141** | **0.897** | **0.791** |

Note:
The best performing data is highlighted in bold.

**Table 6  Results of swapping the Q, K, and V input objects in cross-attention on the Davis dataset.**

|  | Q | K | V | MSE | CI | R$^2$ |
|---|---|---|---|---|---|---|
| Drug | GNN feature | GNN feature | PC feature | **0.223** | **0.896** | **0.726** |
|  | PC feature | PC feature | GNN feature | 0.233 | 0.882 | 0.698 |
|  | PC feature | GNN feature | GNN feature | 0.261 | 0.880 | 0.688 |
| Protein | Transform feature | PC feature | PC feature | **0.223** | **0.896** | **0.726** |
|  | PC feature | Transform feature | Transform feature | 0.231 | 0.885 | 0.702 |
|  | Transform feature | Transform feature | PC feature | 0.233 | 0.881 | 0.695 |

**Note:**
The best performing data is highlighted in bold.

In summary, the experiments confirm that the CAFIE-DTA model maintains high prediction accuracy for both large-scale and small-scale drug–target affinity prediction tasks, surpassing other methods reported in existing literature on key evaluation metrics.

To ensure the reliability and stability of the model, we repeated the experimental process five times to obtain the mean and standard deviation SD. As shown in the Table 4.

## Ablation experiment

To determine the impact of each input part in the model on the results, we conducted ablation experiments on each input part on the Davis and KIBA datasets. Table 5 shows the results of three evaluation metrics for the ablation experimental model and the original model on the Davis and KIBA datasets.

Evaluate the effectiveness of the improvement points of the model through ablation experiments. By comparing the performance of the model on two datasets in the absence of different improvement points, it is demonstrated that the model performs more perfectly with prior knowledge and feature fusion using attention modules. Therefore, the experiment was divided into seven control groups, represented by W/O for the improvement points that have been removed in the CAFIE-DTA model:

(1)-w/o Protein_PC Information: Exclude all physical and chemical information from the drug module in the complete CAFIE-DTA model.

(2)-w/o Drug_PC Information: Exclude all physical and chemical information from the protein module in the complete CAFIE-DTA model.

(3)-w/o MCAT: Remove the cross attention module from the complete CAFIE-DTA model and directly concatenate the feature vectors.

(4)-w/o GNN: Exclude the two-dimensional structural information GNN of the drug portion from the complete CAFIE-DTA model.

(5)-w/o CP-Encoder: Use Transformer Encoder instead of CP Encoder in the complete CAFIE-DTA model.

(6)-w/o protein surface curvature: Exclude the average surface curvature information of proteins from the complete CAFIE-DTA model.

(7)-w/o protein electrostatic potential: Exclude the surface electrostatic potential energy information of proteins from the complete CAFIE-DTA model.

**Table 7  Antiviral drugs among the top 100 drugs with the best affinity predictions for six sars-cov-2 replication-related proteins, as predicted by CAFIE-DTA and based on KIBA scores by MRBDTA.**

| Key proteins in SARS-CoV-2 | CAFIE-DTA | | | MRBDTA | | |
|---|---|---|---|---|---|---|
| | Antiviral drug | KIBA score | Rank out of 3,137 | Antiviral drug | KIBA score | Rank out of 3,137 |
| 1. 3C-like proteinase | Valaciclovir | 14.14881 | 18 | Daclatasvir (BMS-790052) | 13.9089 | 4 |
| | Entecavir | 13.77390 | 41 | Ritonavir | 13.4445 | 21 |
| | Daclatasvir (BMS-790052) | 13.23956 | 82 | | | |
| 2. RNA-dependent RNA polymerase | Daclatasvir (BMS-790052) | 14.33430 | 13 | Daclatasvir (BMS-790052) | 13.4401 | 8 |
| | Valaciclovir | 14.17196 | 21 | Ritonavir | 12.6714 | 29 |
| | Entecavir | 13.64155 | 46 | Entecavir | 12.5049 | 45 |
| 3. Helicase | Valaciclovir | 14.02522 | 26 | Daclatasvir (BMS-790052) | 13.9021 | 3 |
| | Entecavir | 13.78821 | 39 | Ritonavir | 13.3434 | 18 |
| | Daclatasvir (BMS-790052) | 13.58512 | 49 | | | |
| 4. 3′-to-5′ exonuclease | Valaciclovir | 13.99539 | 23 | Daclatasvir (BMS-790052) | 13.7957 | 5 |
| | Entecavir | 13.64906 | 44 | Ritonavir | 13.3427 | 19 |
| | Daclatasvir (BMS-790052) | 13.31367 | 68 | | | |
| 5. EndoRNAse | Valaciclovir | 14.12855 | 19 | Daclatasvir (BMS-790052) | 13.8885 | 4 |
| | Entecavir | 13.730200 | 43 | Ritonavir | 13.4665 | 19 |
| | Daclatasvir (BMS-790052) | 13.166339 | 88 | | | |
| 6. 2′-O-ribose methyltransferase | Valaciclovir | 14.129311 | 20 | Daclatasvir (BMS-790052) | 13.9041 | 4 |
| | Entecavir | 13.779694 | 41 | Ritonavir | 13.4015 | 24 |
| | Daclatasvir (BMS-790052) | 13.118487 | 97 | | | |

From the experimental results in Table 5, it can be seen that CAFIE-DTA is completely superior to the ablation model in three indicators, whether on the Davis dataset or the KIBA dataset. Therefore, the results of ablation implementation indicate that the physical learning information of drugs and proteins helps to improve prediction performance.

Additionally, In order to intuitively reflect the influence of the Q, K, and V input objects we choose in the attention mechanism on the result. The Q and K inputs of the drug are replaced with the corresponding physicochemical information features, and the V input is replaced by the subgraph features. For proteins, we replace the Q and K inputs with the corresponding physicochemical information features, and the V input with the sequence features of the protein. The experimental results in Table 6. Therefore, according to our proposed cross attention input, the model obtained has the best performance.

## Case analysis

Drug repurposing is a novel drug discovery strategy. Firstly, it can significantly shorten the drug development timeline because the safety and pharmacokinetic properties of these drugs have already been verified in their original indications. Secondly, it can reduce development costs, avoiding the expensive and time-consuming toxicity and efficacy tests in early drug development. Additionally, drug repurposing can increase the likelihood of success since existing clinical data provide valuable reference information. By reevaluating existing drugs, potential treatments can be quickly identified, offering faster and safer treatment options for patients. In this case study, we selected SARS-CoV-2 replication-related proteins as targets and applied the trained CAFIE-DTA to predict the binding affinity between 3,137 FDA-approved drugs and SARS-CoV-2 replication-related proteins (*Riva et al., 2020*; *Dittmar et al., 2021*). The purpose of this case study is to provide a real-life application example of CAFIE-DTA and verify its reliable predictive performance in drug design. The FASTA sequences of SARS-CoV-2 replication-related proteins include 3C-like proteinase (accession YP_009725301.1), RNA-dependent RNA polymerase (accession YP_009725307.1), helicase (accession YP_009725308.1), 3′-to-5′ exonuclease (accession YP_009725309.1), endoRNAse (accession YP_009725310.1), and 2′-O-ribose methyltransferase (accession YP_009725311.1). These sequences were obtained from the National Center for Biotechnology Information database (*Zhang, Wang & Chen, 2022*). The binding affinities of 3,137 FDA-approved drugs with six SARS-CoV-2 replication-related proteins predicted by CAFIE-DTA based on KIBA scores can be found in Supplemental File. For each SARS-CoV-2 replication-related protein, CAFIE-DTA was compared with MRBDTA. It was found that among the top 100 drugs with the best affinity predictions, CAFIE-DTA predicted 18 antiviral drugs, while MRBDTA predicted 13 antiviral drugs, as shown in Table 7. Therefore, we found that among the top 100 drugs with the best affinity predictions, CAFIE-DTA was able to predict more antiviral drugs.

MRBDTA only considers the sequence information of drug targets, while CAFIE-DTA not only considers the sequence information of drug targets but also the physicochemical information of drug targets. And these characteristics, such as electrostatic potential energy, hydrophobicity, and the number of hydrogen bond donors/acceptors, are crucial for the specificity and affinity in the molecular recognition process. Secondly, regarding the three-dimensional structure of the target, it reflects the morphological characteristics of the target surface. The specific shape of the target surface can affect how drug molecules approach and bind to the active site.

## CONCLUSIONS

Predicting the binding affinity between proteins and ligands is a crucial and challenging task in drug development. In this study, we propose an innovative model, CAFIE-DTA, which enhances predictions using protein 3D curvature and electrostatic potential information. This model not only considers the physicochemical properties of drugs and proteins but also delves into their sequence and structural features, achieving comprehensive and three-dimensional characterization of both entities. Despite constructing a multi-information, multi-angle cross-attention model to predict

protein-ligand binding affinity, and achieving certain effectiveness, current models still have limitations, particularly in handling physicochemical feature processing, which requires further refinement and optimization. Then, because the binding of proteins and drugs is a dynamic process, the conformational changes can significantly affect the binding affinity between drugs and proteins (*Karplus & McCammon, 2002*). How to accurately obtain the dynamic conformational changes of proteins, and how to use and extract time-series features from these dynamic information, remains a challenging task.

To further explore the practical significance of predicting protein-drug affinity in drug discovery, we selected SARS-CoV-2 replication-related proteins as our study focus to screen potential candidate drugs using the model. Results demonstrate that accurately predicting protein-ligand binding affinity indeed accelerates the drug screening process, paving the way for efficient drug discovery pathways. Lastly, we will continue to optimize methods that integrate physicochemical information, aiming to more fully and efficiently utilize this information to further enhance the model's prediction accuracy and generalization capabilities. Secondly, in order to compensate for the lack of protein dynamic information, we plan to enhance the performance of the model by combining molecular dynamics (MD) simulation information. Future research will obtain more accurate time-series information on protein ligand binding by introducing molecular dynamics simulations based on AMBER software. Adopting and improving the pre training model ProtMD based on protein dynamic information encoding proposed by *Wu et al. (2022a)* for feature extraction of protein conformation in each frame. Thus, the dynamic information of protein ligands is incorporated into the model.

This series of efforts aims to continuously improve protein-drug binding affinity prediction models, providing more precise and efficient tools for drug development, thereby advancing the drug discovery process.

### Funding
The work was supported by the National Natural Science Foundation of China (No. 32470985) and Foreign Youth Talent Program of the Ministry of Science and Technology, China (No. QN2022014011L). The funders had no role in study design, data collection and analysis, decision to publish, or preparation of the manuscript.

### Grant Disclosures
The following grant information was disclosed by the authors:
National Natural Science Foundation of China: 32470985.
Foreign Youth Talent Program of the Ministry of Science and Technology, China: QN2022014011L.

### Competing Interests
The authors declare that they have no competing interests.

## Author Contributions

- Ailu Fei conceived and designed the experiments, performed the experiments, analyzed the data, performed the computation work, prepared figures and/or tables, authored or reviewed drafts of the article, and approved the final draft.
- Yihan Wang conceived and designed the experiments, performed the experiments, prepared figures and/or tables, and approved the final draft.
- Tiantian Ruan performed the experiments, prepared figures and/or tables, and approved the final draft.
- Yekang Zhang analyzed the data, performed the computation work, prepared figures and/or tables, and approved the final draft.
- Min Yao analyzed the data, authored or reviewed drafts of the article, and approved the final draft.
- Li Wang analyzed the data, authored or reviewed drafts of the article, and approved the final draft.

## Data Availability

Code is available in the Supplemental Files.

The resource codes and datasets are available at GitHub:

- https://github.com/NTU-MedAI/CAFIE-DTA.

- Ailu, F. (2025). CAFIE-DTA [Data set]. Zenodo. https://doi.org/10.5281/zenodo.16570534.

The resource Davis and KIBA datasets are available at: https://staff.cs.utu.fi/~aatapa/data/DrugTarget/.

## Supplemental Information

Supplemental information for this article can be found online at http://dx.doi.org/10.7717/peerj-cs.3117#supplemental-information.

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
