# Peer review of "Enhanced information cross-attention fusion for drug–target binding affinity prediction"

_PeerJ Computer Science, doi:10.7717/peerj-cs.3117_

## Round 0.1 · original submission · Major Revisions

Please see the attached review reports.

Reviewer 1 ·

Basic reporting

The paper presents an interesting machine learning model for protein-ligand binding affinity prediction. Benchmark comparisons are conducted using two datasets, Davis and KIBA, demonstrating the model's superiority over existing literature methods. The novelty of the proposed approach lies in its integration of physicochemical information with molecular sequence data. However, several aspects could be improved to enhance the quality of the paper:
1. The manuscript's English could be refined, as some sentences are repetitive in different sections. For example, the discussions on determining electrostatic potential using APBS software and calculating mean curvatures appear multiple times.
2. The descriptions of Model-1 and Model-2 need to be clarified, as the current wording is confusing.

Experimental design

-

Validity of the findings

-

Reviewer 2 ·

Basic reporting

This paper proposes a model called CAFIE-DTA for drug-target binding affinity prediction. The model fuses protein 3D curvature information and electrostatic potential information, and uses a cross-multi-head attention mechanism to fuse physicochemical features and sequence/structure information. Experiments on two benchmark datasets, Davis and KIBA, show that the proposed method has a certain degree of performance improvement over existing methods. Although the paper has made certain contributions, there are still several key issues that need to be addressed.

(1) The introduction does not fully explain the motivation and innovation of the study. The introduction should also briefly mention the related research on enhancing protein 3D curvature information and electrostatic potential information, and its potential role in drug-target affinity prediction.

(2) The literature review in the introduction section describes early work in detail but insufficiently discusses recent research from 2023-2025, particularly the latest approaches based on 3D structural information for drug-target affinity prediction.

(3) Figures 1-6 lack detailed captions and are difficult to understand independently. Figure 1, as the core architecture diagram, needs a more detailed explanation of the input/output and functionality of each module; Figure 6, showing the Delaunay triangulation process, lacks clear correspondence with the main text.

Experimental design

(1) Section 2.2 regarding the acquisition of physicochemical features for drugs and proteins is insufficiently detailed. The authors should specify the exact feature list obtained from PubChem, value ranges, feature normalization methods, missing value handling strategies, and other implementation details.

(2) Section 2.8 contains theoretical descriptions of Delaunay triangulation for calculating protein surface curvature but lacks crucial implementation details such as triangular mesh construction methods, computational efficiency, precision control, etc., making it difficult to reproduce this key step.

(3) The experimental design is somewhat inadequate. The authors should elaborate on data partitioning methods (e.g., k-fold cross-validation), training/validation/test set ratios, model selection strategies, and how fair comparisons were ensured.

(4) The ablation experiments in Table 4 are overly simplistic, only comparing the effect of with/without physicochemical information. A systematic evaluation of each component's contribution (GNN module, CP-Encoder, cross multi-head attention mechanism, etc.) should be conducted, and whether the addition of 3D curvature and electrostatic potential information contributes to model performance.

(5) Section 2.11 briefly mentions training settings (Adam optimizer, learning rate 0.001, etc.) without explaining the rationale for these hyperparameter choices or evaluating their impact on model performance. Hyperparameter sensitivity analysis should be conducted to ensure a reasonable model configuration.

Validity of the findings

(1) The reported performance improvements are relatively small. No standard deviations or confidence intervals are provided, and no statistical significance tests are performed, making it difficult to determine whether these small improvements are reliable and stable.

(2) The SARS-CoV-2 case study has high potential value but lacks in-depth analysis. The authors should discuss in detail the biological significance of the prediction results, compare with experimental validation data, and explain the mechanistic principles of why CAFIE-DTA can predict more antiviral drugs than MRBDTA.

(3) Section "Conclusions " discusses research limitations, but is too general. The authors should specifically analyze technical challenges in handling protein conformational changes, processing macromolecular complexes, considering solvent effects, etc., and their potential impact on model prediction accuracy.

(4) The future directions in the conclusion lack specific planning. The authors should propose clear improvement directions, such as integrating molecular dynamics simulation information, considering protein pocket flexibility, extending to other types of biomacromolecular interaction predictions, etc.

Reviewer 3 ·

Basic reporting

In general, the paper is well-organized. Please see the "Additional comments" for my concerns.

Experimental design

Overall, the paper is technically sound. Please see the "Additional comments" for my concerns.

Validity of the findings

The authors have proposed several interesting modules, notwithstanding a relatively weak improvement. Please see the "Additional comments" for my concerns.

Additional comments

In this paper, the authors proposed a DTA prediction method based on cross-attention mechanisms. Overall, the paper is well-organized and presents several interesting modules like the CP-encoder and triangulation. The authors have conducted experiments to prove its effectiveness and also performed a case analysis. Although the results are relatively fair, the authors released their codes for public reproduction. I have several suggestions to further refine this paper.

1. Please add captions for each figure. For example, the specific explanations for "PCD", "PCP", and "CP" should be given.

2. What does the "CP" in CP-Encoder stand for? Please give its full name.

3. Figure 3 shows that the CP-Encoder module consists of three encoders. Specifically, there are two additional encoders forming as a pair after the residual connection. However, the authors give no explanation regarding here. Why are three encoders introduced here? What is the particular role of each encoder in this structure?

4. Most of the content in section 2.6 is from the paper on the Transformer and is not a new contribution of this work. Please avoid writing too much content that was not originally proposed. In contrast, again, the motivation for using the three-encoder architecture is not mentioned at all. Please strengthen this part.

5. How about exchanging the two features in the cross-attention module? Specifically, using the Transform feature as V, and the PC feature as Q and K.

6. The triangulation for extracting protein chemical features is a key contribution of this paper. However, its particular role is not assessed properly. Please add an ablation variant by replacing the triangulation with the concatenation of those chemical features (not removing it).

7. Similarly, the particular contribution of cross-attention, as another key contribution claimed in this paper, should be assessed via an ablation study.

8. It would be better if the authors could test their models on more datasets, such as the binary classification task.

9. In the Introduction section, the authors ignored another important category for DTA prediction, i.e., the network-based approaches. The authors are advised to cover this branch to reach a comprehensive review. Representative works are as follows: (DOI: 10.1093/bib/bbz147)(DOI: 10.1109/JBHI.2023.3240305)(DOI: 10.3389/fphar.2018.01134)(DOI: 10.1109/JBHI.2023.3334239).

---

## Round 0.2 · Major Revisions

Reviewer 1 ·

Basic reporting

no comment

Experimental design

no comment

Validity of the findings

no comment

Additional comments

no comment

Reviewer 2 ·

Basic reporting

The author has addressed my main concern. However, when introducing other people's work, sometimes it is "name + et al." and sometimes "surname + et al.". I suggest that it be unified. After this minor problem is resolved, the manuscript can be considered for acceptance.

Experimental design

no comment

Validity of the findings

no comment

Additional comments

no comment

Reviewer 3 ·

Basic reporting

The work has been improved.

Experimental design

Some details are ignored. Please see the "additional comments".

Validity of the findings

The findings are acceptable, although the authors refused to validate the performance of their model on the binary classification task.

Additional comments

Despite a significant improvement, the authors did not fully address my concern. Specifically,

1) Please report the detailed results of exchanging the two features in the cross-attention module, rather than simply saying that "the training effect of the model was not as good as the current one".
2) As I have mentioned in my previous comments, the manuscript ignored the network-based approaches in the Introduction section. The authors responded that "Please make specific modifications in lines 111 to 113 of the introduction and highlight them with bright colors", which is quite confusing. Nevertheless, I referred to lines 111-113 but found no network-based methods there. The 3DProtDTA and Subpocket that newly added there in the revised paragraph are clearly 3D methods instead of network based ones. So please cover the network-based category by introducing the following representative works (DOI: 10.1093/bib/bbz147)(DOI: 10.1109/JBHI.2023.3240305)(DOI: 10.3389/fphar.2018.01134)(DOI: 10.1109/JBHI.2023.3334239)(DOI: 10.1093/bioinformatics/btae346).

3) AI-generated response is highly suspected.

---

## Round 0.3 · accepted · Accept

The reviewer seems satisfied with the last edits and changes, and therefore I can recommend this article for acceptance.

Reviewer 3 ·

Basic reporting

It is ok for publication.

Experimental design

The experimental design is solid for publication.

Validity of the findings

The authors have reported their findings clearly.

Additional comments

The authors have addressed all my concerns properly.